# Stakeholder perspectives on an integrated package of care for lower limb disorders caused by podoconiosis, lymphatic filariasis or leprosy: A qualitative study

**Bethany Davies**[1]*, **Mersha Kinfe**[2], **Oumer Ali**[1,2], **Asrat Mengiste**[2], **Abraham Tesfaye**[2], **Mossie Tamiru Wondimeneh**[3], **Gail Davey**[1,4], **Maya Semrau**[1]*, **EnDPoINT Research Team and Consortium**[1,2,4]¶

**1** Centre for Global Health Research, Brighton and Sussex Medical School, Brighton, United Kingdom, **2** Center for Innovative Drug Development and Therapeutic Trials for Africa (CDT-Africa), Addis Ababa University, Addis Ababa, Ethiopia, **3** Ministry of Health, Addis Ababa, Ethiopia, **4** School of Public Health, Addis Ababa University, Addis Ababa, Ethiopia

¶ Membership of EnDPoINT Research Team and Consortium is provided in the acknowledgments
* b.davies@bsms.ac.uk (BD); m.semrau@bsms.ac.uk (MS)

**Data Availability Statement:** Data cannot be shared publicly because we do not have ethical

## Abstract

### Background

Lower limb disorders including lymphoedema create a huge burden for affected persons in their physical and mental health, as well as socioeconomic and psychosocial consequences for them, their families and communities. As routine health services for the integrated management and prevention of lower limb disorders are still lacking, the 'Excellence in Disability Prevention Integrated across Neglected Tropical Diseases' (EnDPoINT) study was implemented to assess the development and delivery of an integrated package of holistic care–including physical health, mental health and psychosocial care–within routine health services for persons with lower limb disorders caused by podoconiosis, lymphatic filariasis and leprosy.

### Methodology/Principal findings

This study was part of the first of three phases within EnDPoINT, involving the development of the integrated care package. Focus group discussions and key informant interviews were undertaken with 34 participants between January–February 2019 in Awi zone, Ethiopia, in order to assess the draft care package's feasibility, acceptability and appropriateness.

Persons affected by lower limb disorders such as lymphoedema experience stigma, exclusion from families, communities and work as well as physical and financial hardship. Beliefs in disease causation inhibit affected persons from accessing care. Ignorance was a barrier for health care providers as well as affected persons. Training and education of affected persons, communities and caregivers is important in improving care access. It also requires time, space, materials and financial resources. Both top-down and grass roots

approval to share data with people outside the research team. Data can be made available to researchers who meet the criteria for access to confidential data, by contacting NTD. DataManager@bsms.ac.uk for this research, subject to approval from the BSMS Institutional Data Access / Ethics Committee.

**Funding:** This study is part of the 'Excellence in Disability Prevention Integrated across Neglected Tropical Diseases' (EnDPoINT) project. The study is supported by the National Institute for Health Research (NIHR) Global Health Research Unit on NTDs at Brighton and Sussex Medical School (BSMS) using Official Development Assistance (ODA) funding (award number 16/136/29). The grant was awarded to GD and is being carried out in partnership between BSMS in the UK and CDT-Africa, Addis Ababa University (AAU) in Ethiopia. The views expressed in this publication are those of the author(s) and not necessarily those of the NIHR or the Department of Health and Social Care. The funder has no role in study design, data collection and analysis, decision to publish, or preparation of the manuscript.

**Competing interests:** The authors have declared that no competing interests exist.

input into service development are key, as well as collaboration across stakeholders including charities, community leaders and "expert patients".

## Conclusions/Significance

This study highlighted the need for the EnDPoINT integrated care package and provided suggestions for solutions according to its three aspects of integrated care (integration into routine care; integration of mental health and psychosocial care; and integration of care across the three diseases), thereby giving support for its feasibility, acceptability and appropriateness.

### Author summary

Lower limb disorders including lymphoedema are prevalent in Ethiopia as a common endpoint of varying causes such as podoconiosis, leprosy and lymphatic filariasis. This study involved the development of a comprehensive integrated and holistic care package for lower limb disorders into routine health care services. It used interviews and focus groups to assess feasibility, acceptability and appropriateness of the draft care package. We found that persons affected by lower limb disorders had many negative experiences due to their condition, especially related to stigma, that included physical, financial and psychological sequelae. Neglect was paramount, with financial neglect in central budgets, a lack of knowledge by care givers and a lack of awareness among affected persons and their communities, all contributing to inadequate care provision and access. Affected persons, communities and caregivers may benefit from provision of learning opportunities about the prevention and treatment of lower limb disorders; and resources are crucial in engendering change, including material goods, time to provide care, and collaborative work to create a culture shift and address stigma.

## Introduction

Lower limb disorders including lymphoedema, resulting from neglected tropical diseases (NTDs) such as lymphatic filariasis (LF), podoconiosis and leprosy, are a substantial problem carried disproportionately by a few highly endemic countries, such as Ethiopia [1]. LF is caused by chronic parasitic infection, leprosy by mycobacterial infection and podoconiosis results from the long-term exposure of bare feet to irritant mineral particles in red clay soil of volcanic origin [2]. Lymphoedema occurs when the lymphatic system sustains damage with impaired drainage of lymphatic fluid. Accumulation of lymphatic fluid in the affected limb causes a chronic progressive swelling. Over time, there is fibrosis and thickening of the subcutaneous tissues. This may be disfiguring, painful and there is a risk of recurrent infections due to the stagnant interstitial fluid [3]. Many of the care needs for established limb lymphoedema are the same regardless of aetiology–for example, a self-care routine, foot hygiene, skin care with emollients, wound care, appropriate footwear and management of acute flares or super-added infections [4].

Globally, it is estimated that there are nearly 15 million cases of LF with lower limb lymphoedema [5], at least 3 million of podoconiosis [6] and just over 200,000 new leprosy cases annually [7]. However, the prevalence and the predominant aetiological cause vary widely

dependent on locality. Ethiopia is comprised of diverse topographical areas with climatically different conditions. Its population is over 93 million, 79% of whom live in rural areas (world bank) [8]. A disproportionate number of its population suffers from lymphoedema. Mapping by Deribe et al in Ethiopia in 2015 found that 6.2% of their study population of 129,959 individuals across the country were living with lymphoedema [9]. In contrast to the relative proportion globally, podoconiosis is responsible for over 60% of lymphoedema cases in Ethiopia. However, there is also significant geographical variation within Ethiopia itself, as prevalence varies between health districts from zero to 8.6% [9]. Three hundred and forty-five (of 839) *woredas* are endemic for podoconiosis; 112 for LF; and of these, 53 are co-endemic for podoconiosis and LF [6].

The World Health Organisation (WHO) launched the Global Programme to Eliminate Lymphatic Filariasis in 2000, with a 2011 position statement on morbidity management and disability prevention (MMDP) [10]. This recognized that, in addition to the interruption of disease transmission, the management of morbidity and prevention of disability is a core component of care. To improve quality of life and alleviate suffering, WHO recommends a basic package of care for affected persons, including ongoing access to continuing care and support. Although the position statement urges members to consider a combined approach with other disease-specific programmes tackling similar chronic disease, the toolkit focuses on LF rather than lower limb disorders more generally. This is problematic in an area where LF is only one of several possible aetiologies. In areas where LF, leprosy and podoconiosis are all endemic such as in many parts of Ethiopia, differentiating between causation is not straightforward and so a care package that incorporates all three NTDs may be more useful.

Ethiopia has had foot care for people affected by leprosy embedded into routine care since 2001, but services for podoconiosis or LF related disease are fragmented and only patchily provided by external contributors [6]. In contrast, the Ethiopian Federal Ministry of Health (FMOH) developed integrated guidelines for MMDP related to podoconiosis and LF. However, these omit leprosy, and neglect any mental health or psychosocial care. Recognizing that there can be overlaps between the two terms, here we take mental health care to mean services that aim to improve a person's psychological or emotional wellbeing, for instance interventions aiming to reduce depression or anxiety; we take psychosocial care to mean interventions that explicitly take into account interrelating social and individual factors–stigma reduction interventions could be an example of this. Realizing the opportunities afforded by integration (i.e. integration, 1) across diseases; 2) into routine health care; and 3) incorporating mental health and psychosocial interventions), the Ethiopian FMOH has identified podoconiosis and LF as priority NTDs and included them in the first two National NTD Master Plans (2013–15 & 2016–20) and within the Health Sector Transformation Plan (HTSP) 2015/2016–2019/2020. These are moving away from disease-specific vertical programmes (as are currently implemented as part of the national health structure, and which have fallen out of favour to a degree due to their lack of integration, lack of efficacy and potential for harm) [11] and towards cross-cutting interventions integrated with non-NTD programmes.

The current healthcare system in Ethiopia consists of a mixture of public, private and non-governmental organisations. The public health strategy is for fair access to health services for all, with a tiered structure of service provision, from specialised hospitals (regional/general hospitals and primary/district hospitals), through to primary health care units (health centres with satellite health posts). The health service extension programme was introduced in 2003 to bring "high-impact primary care and community-based" healthcare into households, especially in rural areas, countering the previous imbalance and neglect [12]. Before the reforms, Ethiopia was not only in the lowest quintile of African nations for healthcare workers (HCW) per population, but these were unevenly distributed in terms of number and skills, with major

gaps in essential services in rural areas, which carry the greatest burden of lymphoedema and its consequences [12].

The study is part of Phase 1 of the 'Excellence in Disability Prevention Integrated across NTDs' (EnDPoINT) implementation research study, whose overall aim was to develop and implement an integrated, comprehensive and holistic care package–including physical health, mental health and psychosocial care–for lower limb disorders across the three diseases into routine health services in Awi zone in North-West Ethiopia. This particular study aimed to explore the perspectives of a range of stakeholders to harness their views on the feasibility, acceptability and appropriateness of such integrated care. For the purposes of this study: feasibility is conceptualized as the capability for successful completion of the project; acceptability, as pre-intervention prospective treatment acceptability [13]; and appropriateness, the suitability of the project for the needs identified.

## Methods

### Ethics statement

Ethical approval was obtained from the Brighton and Sussex Medical School Research Governance and Ethics Committee, UK (ref. ER/BSMS9D79/1) and the Institutional Review Board of the College of Health Sciences at Addis Ababa University, Ethiopia (ref. 061/18/CDT). All participants were given a participant information sheet (PIS) in Amharic and were required to give written informed consent before taking part, if literacy allowed. If they were unable to read the PIS or sign the consent form due to illiteracy, then the form was read out, and they were asked to give witnessed oral consent. Participants were able to withdraw from the study at any time without detriment or reason up until the point at which data had been aggregated. In the event that any participant was deemed to need immediate assistance or support, for instance because they disclosed mental distress, appropriate action was established according to local resources and capacities.

### Study setting

The EnDPoINT study was conducted in three districts (of 12) in Awi zone, Amhara Region, north-west Ethiopia. The Awi zone in the central highlands of Amhara has the red clay soil, seasonal rainfall and altitude that are necessary for the development of podoconiosis,[14] and 87.5% of its population live rurally [15]. The EnDPoINT research team together with the Ethiopian FMOH selected the particular districts for the study based on a combination of their accessibility, previous or ongoing work in the area and their co-endemicity of podoconiosis, LF and leprosy. The pilot study was conducted in Guagusa Shikudad *woreda* (district).

### Study design

This study was guided by both the Medical Research Council (MRC) framework for complex interventions and the Context and Implementation of Complex Interventions (CICI) framework [16,17], building through three phases with an iterative approach, and with "Theory of Change" embedded for the development and feasibility/piloting phases [18]. Further details on these frameworks and how the EnDPoINT project maps onto them are provided elsewhere [19].

Phase 1 of EnDPoINT, which this study was part of, entailed a number of research activities to inform the development of the holistic care package. This care package was based on an established MMDP (physical) self-care package for podoconiosis and LF, but building in mental health and psychosocial components, and including leprosy care for lower limb disorders.

This paper reports on one element of these Phase 1 activities: key informant interviews and focus group discussions with stakeholders. These were intended to assess feasibility, acceptability and appropriateness of the draft care package as viewed by the stakeholders, as well as assessing key aspects of the Theory of Change, such as assumptions made, for example the willingness and availability of key stakeholders to participate in and engage with the various care package interventions.

Three focus group discussions (FGD) and 11 individual key informant interviews (KII) were conducted in January and February 2019 by researchers MK, OA and AM, involving 34 participants in total (25 male and 9 female). Three participant groups were recruited: regional health office NTD staff, staff members at Injibara hospital working on NTDs, and community members including affected persons. Purposive sampling with snowballing was used to identify and recruit key stakeholders based on their role/position in the community; the sample size was guided by the number of key stakeholders who were identified to take part and available, as per the original study protocol [19]. Participants had to be at least 18 years of age.

Each of the three focus groups were separated according to a different stakeholder group, as follows: 1) community representatives (*kebele* administrators, religious leaders, affected persons, health extension workers); 2) health professionals (nurses, health officers, general practitioners and specialists); and 3) decision-makers (NTD focal, NTD officer, TB-Leprosy officer, programme managers). There were three affected persons (two female and one male) in the community representative groups. The sample size during focus groups was limited to ensure that all participants have the opportunity to speak.

The KIIs included participants from the FMOH (officers for leprosy and podoconiosis), staff members from International Orthodox Christian Charities (IOCC), a dermatologist from Felegehiwot hospital, and several focal team members from Awi zone (NTD focal, leprosy focal, mental health focal), staff from the Guagusa Shikudad health office, the head and vice head of the *woreda* office, as well as the *woreda* NTD focal person.

Open-ended questions with follow-up clarification and further enquiry were used [see S1 Text for the topic guide]. Questions were based around participants' experiences regarding affected persons with lower limb disorders secondary to podoconiosis, LF or leprosy. All participants were encouraged to contribute and be heard; the discussion facilitators set ground rules for behaviour to support participants to feel safe; they also actively tried to include all participants in the discussion and to make the participants feel their contribution was welcome and worthwhile. Those participants who were not highly verbal were encouraged to share their views. The interviews and focus groups were conducted in Amharic, audio recorded, transcribed and checked for accuracy. Unique identifying numbers were assigned during transcription, and names of specific people removed if mentioned. A meaning-based translation of transcripts into English was made before coding.

## Data analysis

Data were thematically analysed using a combination of manual coding and nVivo software. Two researchers (BD and MK) independently reviewed the data from the focus groups and the first few interviews initially, ensuring detailed knowledge before using within-case analysis, close text analysis and cross-case analysis to develop initial themes. These were then compared and integrated to form a coding framework which was then applied to the remainder of the data. Examples of disconfirming evidence were also sought, as well as comparing responses about different stakeholders. Data collection continued until no new themes emerged.

Confidentiality and anonymity of data were ensured by using allocated identifiers instead of names, and with identifying data kept separately. Data were fully anonymized before data

analysis. Data were stored in a secure OneDrive folder with access limited to members of the study team for whom it was essential.

Quotations below are identified as follows: Key informant interviews were labelled K1 – K11 in chronological order of their interviews; focus group participants are identified by the focus group attended and their allocated number at the start of the session. For example, [P2, FG3] was participant number two in the third focus group.

## Results

In total, 34 adults between 24 and 52 years of age participated in focus groups and individual interviews. The majority of the participants were male (74%).

Overall, the focus group discussions and key informant interviews highlighted the need for the EnDPoINT integrated care package and provided suggestions for how it might provide solutions according to its three aspects of integrated care (integration into routine care; integration of mental health and psychosocial care; and integration of care across the three diseases), thereby giving support for its feasibility, acceptability and appropriateness. Table 1 and Fig 1 summarise the main themes and subthemes identified.

### Establishing a need for the new integrated care package

**Disease burden and its consequences.** The **magnitude of the problem** in the study districts due to the numbers of affected individuals with lower limb disorders was described:

*"It was one [] disease that the community is suffering a lot with"* (FG1, P2, 59-year-old male community representative)

*"We have treated fifty thousand, it is entered in the data base, and the Ministry of Health also knows about this [. . .], there are these many people but we only addressed about fifteen and sixteen woredas."* (K4)

Even then, these numbers did not reflect the true extent of disease burden, due to cases that had not yet been identified or brought into services.

*"There are so many people. . . more than 500 people have come to get the service. And there are also people with the problem who didn't come to the health facility yet."* (K8).

Participants described the **consequences borne by individuals and families** as a result of living with lymphoedema in Ethiopia, *"having this problem and living in sorrow"* (FG1, P1, 48-year-old female community representative).

Three main categories were offered: **physical, psychosocial and financial**:

*"Physical impairment brings social impact that leads to psychological impairment"* (FG3, P2, 37-year-old male decision-maker).

The predominant **physical burden** resulted from **limb swelling restricting mobility**:

*". . .they cannot wear a shoe and walk freely because of the swollen limb/leg. Even the size of the shoes didn't fit their leg. They are suffering a lot because of this."* (FG1, P5, 52-year-old male community representative).

For many there was an additional burden attributed to an **offensive smell**:

**Table 1. Themes and sub-themes identified through qualitative analyses of focus group discussions (FGDs) and key informant interviews (KIIs).**

| High level themes | Sub-themes | |
|---|---|---|
| **Establishing a need for the new integrated care package** | The disease burden in the community and its consequences | The size of the problem: the large number of persons affected with lower limb disorders, as well as a need to identify those affected |
| | | The financial consequences on affected persons and their dependents |
| | | The physical burden of disease on an individual: mobility, smell, swelling |
| | | The psychosocial effects of disease: stigma–affected persons, families, community and healthcare providers; effects on intimate relationships, immediate family and exclusion from key social events |
| | Current and anticipated challenges and barriers to care | Distance as a problem: the geographical spread of disease and affected persons |
| | | Knowledge deficit: a lack of knowledge among healthcare workers, affected persons and community, inhibiting care and leading to stigma. |
| | | Causation beliefs: religious, contagious, hereditary |
| | | Neglect: lack of treatment provision in current model of care, including space, budget, time to provide care and staff turnover |
| **Solutions** | | |
| **Integration into routine care** | Training needs | Community and affected persons: expert patients, demonstrable change, what constitutes care |
| | | Healthcare workers |
| | Material matters | Sourcing and funding material goods: shoes including fit, clean water, perception of materials |
| | Space matters | Separate room to provide care: prioritizing to enhance engagement, stigma between clinic patients |
| | Time as a resource | Long consultations |
| | Who contributes | Cascade of care, external organizations, patient associations, community leaders, self-care: collaborative approach, top down or bottom up |
| **Integration of mental health and psychosocial care** | | |
| **Integration of care across the diseases** | | |

"*People didn't consider them even as a human being and make them sit away from the gathering because they think they have a bad smell.*" (FG1, P2, 59-year-old male community representative).

Participants reported that affected persons **struggle financially** as a result of their limb disorders. Physical impairment limits their ability to work as they had formerly: "*There is a high chance, because of their disability, that they face economic problems and go out to the streets to beg.*" (K3). Farmers especially were disadvantaged as the work predisposes to disease, and disease prevents physical work such as farming: "T*he disease itself banned them from their farm*" (K4). There was also the indirect contribution of stigma and discrimination: "*Some of them can't even work because of the severe injury or because of the social stigma*" (K11). In addition to the financial difficulties resulting from the effects of disease on the ability to work and earn, the conditions themselves incurred additional costs:

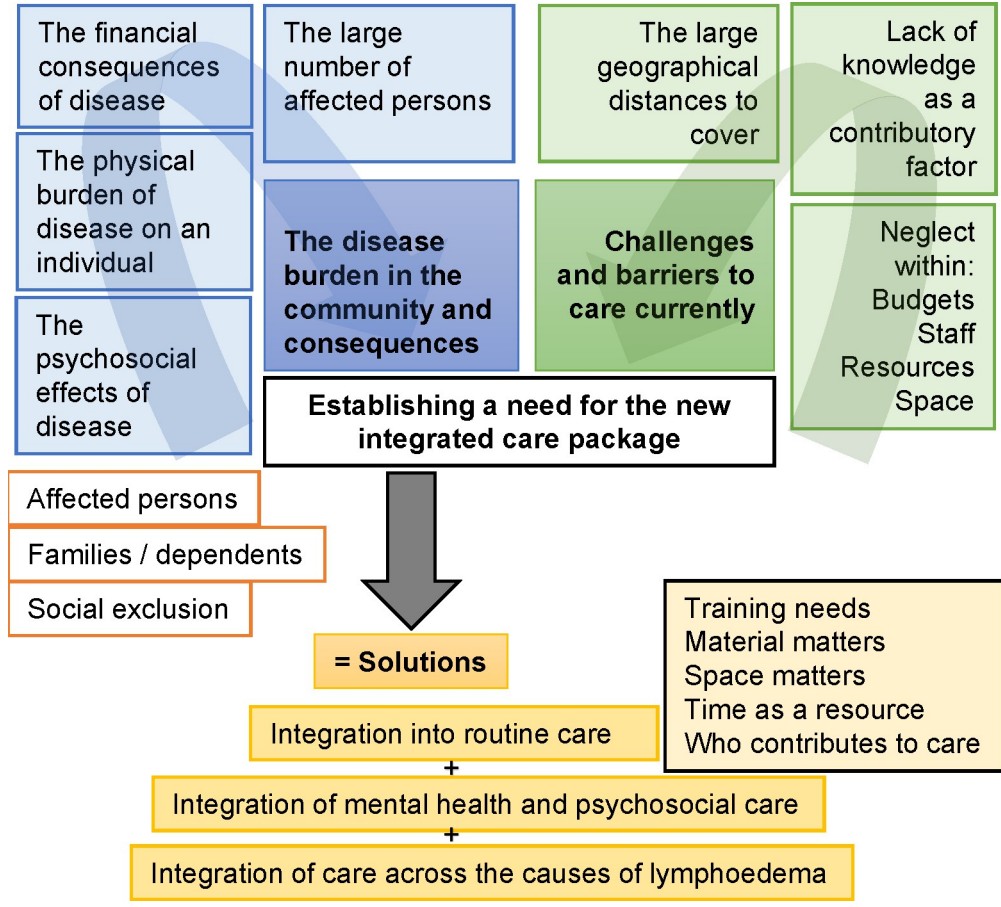

**Fig 1. Themes and sub-themes identified through qualitative analyses of focus group discussions (FGDs) and key informant interviews (KIIs).**

*"Health insurance, that small amount of 200 [roughly US$5], there were those who were not able to pay that amount; the reason is the disease itself"* (K4)

Participants reported **stigma** in all aspects of an affected person's life:

*"Due to their degree of disability, they feel ashamed/ embarrassed and such feelings to eat with other people, to get medical record and proper treatment, to express their views in the proper way like any other person."* (K3).

This is enacted by themselves, their families, their communities and the health professionals providing their care:

*"The stigma comes from different angles, including first from health care providers, second caregivers including family and community and also it's observed that the patients stigmatize themselves."* (FG3, P2, 37-year-old male decision-maker)

Participants described how affected persons felt "ashamed", stay "hidden" and "isolate" themselves, and of a "feeling of inferiority". **Close personal relations** were affected, with disease leading to divorce or preventing marriage in the first place: *"We found divorced husband*

*and wives because of the diseases*" (K4). They were excluded from **key social events**, such as weddings, funerals and other religious services: "*They can't go anywhere, even to express their condolences at the mourning ceremonies.*" (FG1, P2, 59-year-old male community representative). Communities contribute to the discrimination and isolation of individuals: "*Most of the time they get social rejection from the community and from their neighbours. This made them psychologically tortured.*" (K11). This impacted not only on the affected person themselves, but also on their **dependents**: "*Finally, these kids start to decrease with their grades because of the social stigma.*" (FG2, P8, 38-year-old male health professional). Stigma also negatively affected physical health in addition to mental health, with individuals reportedly not engaging with healthcare due to the stigma and shame that they felt: "*Most patients coming to health institutions for services raised stigma as main reason for not seeking health care.*" (FG3, P3, 39-year-old male decision-maker).

**Challenges and barriers to care.   Distance as a problem**: the physical size of the country and the current infrastructure (both in transport and within healthcare) was recognised as potentially problematic,

"*We must think about reaching the community for treatment because as you see it is a big country.*" (FG2, P6, 34-year-old male health professional);

"*it is difficult for health professionals to reach them.*" (FG3, P7, 49-year-old male decision-maker).

The spatial distribution of affected persons across large geographical areas was noted to be especially difficult (although not insurmountable) for those with lower limb disorders for whom mobility may be restricted, painful or even harmful:

"*It is not common to treat these chronic cases or such cases at hospital or centralized hospital or referral hospital, for those cases a long journey is not appropriate—to move that long way that may cause some harm to them.*" (FG3, P7, 49-year-old male decision-maker).

**Knowledge deficit**: A lack of knowledge or awareness around lower limb disorders was reported by a wide range of participants, including affected persons, families, communities, care managers and caregivers. There was widespread misunderstanding of the **underlying causes of the disease process** among affected persons, their families and wider society. Not only was a lack of knowledge noted, "*There is still a knowledge gap in the community about the causes of the disease and the ways of disease transmission.*" (K5); but beliefs ascribing causation to religious, contagious or social practices were reported to be widely held which actively hindered the engagement of affected persons and their families with medical care:

"*They believe only God can heal them but no other treatment. . .If they go to the health facilities, they could get better but they don't believe. They think it is given by God and cannot be treated in the health facility.*" (FG1, P3, 36-year-old female community representative);

"*Traditionally, the community assumes the disease is hereditary or a curse from God. This hinders them from treatment in the health facilities.*" (K1)

"*The main challenge in the community is the perception of people to the disease itself. They perceived it as a curse from God and not a treat[able] disease.*" (K9);

"*They only think it is because of their genes and did not think of getting better through treatment.*" (FG1, P2, 59-year-old male community representative)

**Neglect:** These diseases were also neglected within all levels of the health care system:

*"It is a very important issue but no attention was given before."* (K8);

*"In the meetings we hold, they say they are forgotten by the government and are denied attention, these issues are raised a lot [. . .] it is also a problem I witnessed in practice."* (K3)

This lack of attention or awareness manifested for affected persons as **a lack of adequate healthcare provision**, *"There was no treatment for the disease neither in the health facilities nor in the health centre until now."* (FG1, P5 52-year-old male community representative) Healthcare workers were insufficiently aware or trained to deal with such cases, *"People with podoconiosis might come to the OPD [outpatient department] but the health workers didn't give attention for it, they would rather assess and treat other cases and let the patient go by completely ignoring the podoconiosis."* (K7). If staff were trained, then **turnover** was frequently cited as a problem: *"The other challenge is staff turnover. Those who have already trained and who have the skill to provide the service also leave the health facility and go to other places. So, this is a major problem."* (K5) This appeared to be a widespread problem across the participants' areas of work. One participant ascribed it to a low salary, but the majority did not expound on the reasons for this turnover, nor on ways to address it, rather on its consequences alone:

*"He or she may be the only professional person in that place. So, once he or she transfers, it causes a problem."* (FG3, P4, 38-year-old male decision-maker)

It was also difficult to establish **space to provide care** for those who were keen to do so:

*"When we started work at the beginning around 2010, even in Amhara region, health centres were not happy to give us a working place."* (K4)

The lack of attention within the health system also featured within **budget planning** and financial resources. Participants involved in health centre services reported struggling with limited financial resources to maintain adequate supplies and carers:

*"There are so many things which we don't achieve from our plans because [. . .] there is no finance."* (K3)

## Solutions: integration into routine care

**Training needs.**   Engaging people with greater understanding and awareness of the diseases through training and education was felt to be essential.

*"The next thing is to change the old perception of the health professionals at the health centre level and to give treatment. In addition to this the religious leaders, the health extension workers and those influential community leaders need to teach the society so the community accept these people."* (K1)

A culture shift was described as a key goal, both within the community and in healthcare services, *"Changing the mind of the society is what needs to be done."* (P2, FG1). Participants for the most part thought that there would be a willingness to engage with such an agenda rather than "resistance", *"I also believe the community is willing too"* (FG2, P2, 59-year-old male community representative).

Raising awareness was thought to be of benefit through enhanced care-seeking and improved self-care, "*If we teach the community changes can happen.*" (K1) **Affected persons** would be more likely to seek care once they were aware that there are services and effective treatment available: "*By giving community awareness to go to health institutions and believe [they can] be cured and that the management works.*" (FG3, P3, 39-year-old male decision-maker). Once affected persons had been registered and engaged with available services, they would also benefit from understanding more about their underlying disease process, why self-care measures are important and how to manage self-care: "*They need to know what causes the problem, how to prevent it and how to treat it. If we can teach them that they can get back to their previous health condition, I think they can get rid of the problem.*" (K8). A wider knowledge of causation and prevention measures was also thought to be required: "*When we talk about control and management of these diseases, the first thing we need to focus on is on the prevention. Because prevention is better way of management.*" (FG2, P5, 30-year-old male health professional). Snowballing of awareness as it may be shared onwards was recognised to have beneficial effects: "*if the community understand it clearly, they will also teach others.*" (FG3, P1, 32-year-old male decision-maker)

Many participants reported that **"expert patients"** (i.e. individuals who themselves had previously been successfully treated for lower limb disorders) would be an invaluable source of teaching and information for new or current patients. They recognised that such people may have a better appreciation and understanding of the problems encountered in living with these conditions. This allows a better shared understanding with affected persons still learning about their condition: "*Use people who were cured as a result of your project. They can teach the before and after changes and this may help others to understand the treatment mechanism in a better way.*" (K4). They would also act as positive role models, advocating the gains and positive outcomes of treatment, "*To bring others, those with good management outcome works by giving their testimony at some Woredas*" (FG3, P3, 39-year-old male decision-maker)

Some participants felt that there would be better understanding and uptake of measures across groups if they were able to see a **demonstrable change**. As with the expert patients, the reinforcement for both healthcare workers and affected persons in being able to see a positive outcome at first hand was thought to have a more powerful effect on their engagement with the program.

> "*The community cannot accept it only because they are told. However, with the awareness we create, if we involve them in jobs and when they can become independent and change their lives, that is when we can make the community believe and accept it.*" (K3).

Equally, there were concerns that an absence of any demonstrable effect could be detrimental. Along similar lines, healthcare workers identified that patient perceptions of **what constitutes care** may impede their engagement. The simplicity of the tools and the actions needed for successful care, such as washing their leg, and wearing shoes meant that patients may not accept that that was all that was required,

> "*They expect injection and when you tell them that washing is the method that we use to treat their disease, they will not accept it.*" (K4).

Shifting that mindset was an important part of engaging people with their treatment, "*we have to convince them that it is medical treatment.*" (FG3, P1, 32-year-old male decision-maker)

A lack of good working knowledge among healthcare workers also needed to be addressed to enable quality care to be delivered. **Training of the healthcare workers** was repeatedly brought up as an absolute necessity,

> "*Because one's knowledge won't be enough unless someone else with better knowledge trains. That way I could get to work easily. But if there is no one to train me, I would be useless.*" (FG2, P2, 48-year-old male health professional)

**Health extension workers** were identified as having a key role in disseminating awareness at a community level, identifying cases and engaging them with care: "*if we can capacitate the health care providers and if we can create awareness up to the level of health extension workers, there will be increased health seeking behaviour among people who are affected by these problems.*" (K6). They reported making efforts to identify all cases within their care remit, both to establish the true extent of disease burden as well as to ensure individual patients were encouraged and supported to engage in care: "*The need of the patients comes next after seeing how many they are in number.*" (K11)

**Material matters.** Participants were clear about **essential material resources** and their conspicuous absence in the delivery of care to people with lower limb disorders. Shoes, medicines, and moisturisers were most commonly identified, "*The most important thing for these three diseases is pair of shoes.*" (K10). **Clean water** was also raised as a key rate-limiting step for delivering care: "*If you say to them to wash at home and if you ask them the availability of clean water, most of the times their response is clean water is not there.*" (K4). The current situation in obtaining material goods to provide care, including the funding needed as well as secure supply chains was thought inadequate. **Shoes** in particular were a cause for concern. The main issues arising were where appropriate footwear could be sourced (in addition to who would cover the costs), and of the additional difficulties posed by the need for shoes to actually fit–one size does not fit all:

> "*Lack of shoes, so this is a common problem everywhere. Because of their unique size and shape, there is a shortage of access for shoes for Podo, LF and Leprosy patients.*" (FG3, P4, 38-year-old male decision-maker)

As well as the material goods themselves, a change in affected persons' approach to the items was seen to be important, **shifting their perception** from this being an everyday item, **to considering it as a treatment**. Achieving this would support affected persons in procuring and safeguarding these resources themselves:

> "*The soap and the basin, there is a tendency to use these materials for something else. When you give them these supplies, you have to tell them not to use the materials for other purposes and you should tell them to consider these supplies as medication*" (K4)

**Space matters.** **Space and place** were recognised as being an important component of care. An increase in the number of affected persons receiving care necessitates an expansion in the spaces available to deliver that care. However, this did not seem to be an obstacle that concerned participants: "*There is no problem regarding the room;*" (K2). Providing care in the affected person's own home was a helpful factor in this regard, "*They did this in their house so for working place we didn't expect to have that much challenge.*" (K2)

Participants were also keen that services at the treatment centres should be separated out from mainstream care, "*If we are integrating these three cases together then we need to assign a*

*specific room for a better work.*" (FG2, P5, 30-year-old male health professional). They predicted that with the burden already carried by these individuals, they may be more likely to engage with care and to have a more positive experience with separate care facilities.

> "*I thought there will be discomfort among patients if they assigned in a same [general] OPD. These patients may feel ashamed even when they sit together with other patients. So in my opinion it is better for them to have a different treatment room.*" (FG1, P4, 46-year-old female community representative)

> "*It helps them to discuss their problems freely with the health care givers which could give them some relief.*" (FG1, P5, 52-year-old male community representative)

A **separate facility** with trained staff could provide benefits in affected persons being prioritised both physically and psychologically.

> "*It is good to give them priority and a chance to get the treatment first.*" (FG1, P5, 52-year-old male community representative)

> "*There is a heavy patient flow in the health centre as you all know. If we make these people wait for the service as others do in OPD, sometimes they might not get the care they want and go back to their homes. Because of the long list of cards for a day in our health centre, there are some patients who could not get treatment service. If these people are left to sit and wait for the line in OPD they might not come back again. So like the ART [HIV service] clinic clients must get the service they needed any time they came and every person knows from where to get the treatment he needed. They won't get frustrated and the service is active. They are properly treated.*" (FG1, P4, 46-year-old female community representative)

**Time as a resource.** These are complex conditions and even with training, consultations may not be straightforward or quick:

> "*In the others case you will sit in your outpatient department and treat your patient. You have a prescription and you ask your patient what his illness is, whether it is his foot and then you just write for him and tell him to go and buy the medication, but in the cases of these diseases it is not something like that. You might be expected to sit down and demonstrate on how to wash their leg. [. . .] you won't just write down and say the patient, go: you put a basin and with water wash your leg and then you do it at home and use Vaseline and do like this and then put on a shoe. If you say the patient to do in this way and if you write it down and give him, he won't do it.*" (K4)

**Time to provide appropriate care** was also a concern for an already busy care team,

> "*The main concern providers raised during the training was that they have no time and are busy with other competing priorities.*" (FG3, P3, 39-year-old male decision-maker).

Improving care for people living with lower limb disorders was seen to need more personnel with the corresponding time to provide care once trained: "*The biggest struggle in the first place is human resource.*" (K1). However, others disagreed and were more confident that this would not be a barrier: "*In regard to the manpower also we don't have a problem.*" (K9)

**Who contributes to care.**   The governmental health system is central to care delivery, with its tiered structure facilitating **cascades of care**, "*we want them to be addressed by the available structure of the health system.*" (K6). This allows access up towards increasingly specialised services:

> "*The trained professionals work in areas where cases are presented. If they find any case that is beyond their and the health centre's capacity, they immediately refer the cases to the hospital because there are more professionals with advanced material for treatment and good knowledge.*" (K11)

and out to wide-reaching disseminated local care. This was thought to allow for the greatest number of affected persons to be reached, while maintaining access to specialist care when needed.

> "*When we implement it directly by ourselves, we couldn't cover large areas but when we change our strategy and gave the responsibility to the health centres after we gave them appropriate training, we could cover very large areas.*" (K4)

Participants felt that a **top-down approach** to initiate and embed widespread change is necessary: "*I think direction should be given at a national level*" (K7); "*I believe that the ministers' attention for the program is a major input to control and eliminate these diseases on the anticipated period of time.*" (K6). Additionally, they felt there was benefit to feeding in the experience and knowledge of those actually receiving and delivering care to improve services: "*The ones who are more knowledgeable about the program are the experts who are implementing the program. Because, they know how the program can benefit the community, how to address the community and how the program can be strengthened.*" (K5)

**"Collaboration"** between stakeholders allows for better outcomes–heightened awareness, sharing of resources and sustainability of services: "*For the sake of the community, the zonal health department, the regional health bureau and the woreda health office should work in collaboration.*" (K1)

Other **external organisations** are essential contributors to initiating, establishing and sustaining a successful program for improved healthcare services for people with lower limb disorders. In addition to raising awareness and instigating change, partner organisations provide training to carers and affected persons: "*The IOCC in collaboration with the Health Bureau are providing training.*" (FG3, P2, 37-year-old male decision-maker); as well as material resources: "*the first thing the Carter Centre did was giving MDA [mass drug administration] for lymphatic filariasis.*" (K1).

**Religious and community leaders** work to raise awareness and are seen as being "influential" within society: "*Even at the grass roots level, influential peoples like leaders, religious leader's engagement would be the best way to bring change at community level.*" (K9). They share information on prevention and treatment, "*We religious leaders were also teaching and counselling them to take medical treatment and to wear shoes.*" (FG1, P2, 59-year-old male community representative), as well as work towards shifting attitudes towards acceptance and away from stigma: "*The religious leader, the health extension workers and those influential community leaders need to teach the society so that the community give acceptance for these people.*" (K1)

**Patient associations** are not regarded so universally as positive. There is an advocacy role, "*it will help their voice [be] heard and protect their rights*" (FG3, P1, 32-year-old male decision-maker), and more directly for affected persons, helping them to establish links with healthcare

services as well as with other people living with the same condition: "*They help each other out and disseminate useful information easily.*" (K10). However, associations may struggle due to internal and external factors: "*The existing associations are not strong enough and not well functioning*" (K9), "*There are situations where one sees the other as a rival*" (K3). Also for the individual, costs of joining the associations [10 Ethiopian Birr / US $0.22 per person] could be prohibitive; "*When they told us how much to pay for being a member of the association it was shocking.*" (FG2, P1, 33-year-old male health professional) and affected persons outside the associations may be excluded from benefits meant for all.

Finally, affected persons are expected to participate actively in their treatment and recovery, once they have been taught how to do so, through ongoing **self-care**: "*the lifetime care is done by the patients themselves.*" (K3)

## Solutions: integration of mental health and psychosocial care

Historically, routine care has not included psychological aspects of care: "*The social, psychological, physical and financial therapy is somewhat forgotten. The biggest gap of the country for not achieving the 2020 goal is also these.*" (K10). This lack of psychological care was viewed as a critical deficit, due to an intrinsic link: "*Physical impairment brings social impact that leads to psychological impairment. These are linked and working together is very important and effective.*" (FG3, P2, 37-year-old male decision-maker). This applies to all three diseases in the integrated care package: "*Stigma is so common and bad among all the three diseases*" (FG3, P2, 37-year-old male decision-maker); "*It is clear that the three diseases need psychosocial support.*" (FG3, P7, 49-year-old male decision-maker). Participants supported combining psychosocial care with routine medical care: "*While doing the integrated treatment of those three diseases it will be even better to include mental health on it because mental health is mostly neglected.*" (FG2, P8, 38-year-old male health professional)

Participants thought that this could be one way of better addressing the problems experienced due to stigma,

> "*They need to have psychological support in order to avoid the stigma and discrimination they face due to their physical disability.*" (K5)

> "*The disease by itself can bring social discrimination. But counselling and developing their mental capacity during rehabilitation sessions can be effective. So this integration idea is a good opportunity for policy makers.*" (FG2, P4, 36-year-old female health professional)

Other ideas on how to specifically address stigma mirror the themes already discussed: collaborative work, "*We can control the stigma and discrimination if all of us including those from different sectors like health centres, woreda office or hospitals can take full commitment.*" (FG2, P7, 33-year-old female health professional); involving influential community leaders, "*if we can use schools and religious organizations*" (K8); addressing the knowledge gap: "*the stigma and discrimination will change if their knowledge about the thing is advanced.*" (K11) and using expert patients: "*It is better if those who are cured and saved give witness and awareness about stigma on the stage face to face to the people.*" (FG3, P7, 49-year-old male decision-maker).

## Solutions: integration of care across the causes of lower limb disorders

The EnDPoINT care package was seen as a way to improve the services delivered for people living with lower limb disorders and consequently their outcomes. These would benefit the affected persons themselves, their families, and wider society. "*People were isolated then, we brought them [. . .], and after six months they came to know the disease is not communicable and*

they [are] able to have family, close friends and even life partners." (FG3, P2, 37-year-old male decision-maker). Healthcare providers also gained through professional satisfaction in achieving good care outcomes: "*Some providers [who] get [made] aware and start treatment and management, [and] start seeing improvement within three months are considering their jobs as blessings.*" (FG3, P5, 42-year-old male decision-maker)

The majority of participants were in favour of integrating services across the three diseases due to the shared elements of the disease and treatment process:

"*Though there are different positive agents of the three diseases they have many things in common. For example, the three diseases can result in disability, foot care can reduce their burden, all are prone to stigmatization. Therefore, providing integrated foot care is very important*" (FG3, P4, 38-year-old male decision-maker)

"*Both lymphatic filariasis and podoconiosis manifest with leg swelling and the case management is similar. That means it's necessary to have integrated management.*" (K6)

An integrated service may be easier for people living with lower limb disorders in **gaining initial access** to foot care services, at a stage when they may not know what is causing their disease: "*We told them to accept all cases of foot swelling [. . .] if the cases are not LF or Podo, we will not tell the patients to return to their home but the health professional will give the treatment that he can.*" (K4); and may encourage greater engagement: "*If three of them are integrated, the health seeking behaviour of the community will also increase.*" (K11); "*when we are able to give common service for those people without differentiating one from another*" (K2).

Most participants thought that integration would allow a more **efficient and cost-effective** service: "*There is nothing to lose by working in an integrated way. We can reach the community with an integrated package of services, this will improve staff efficacy, reduce wastage and redundancy. Generally, for me it is a great way.*" (K9). "*If we apply these packages together, this is a process by which we can bring big changes. Leprosy, podoconiosis and lymphatic filariasis are foot-related problems, therefore, by providing more integrated service, there is the possibility that integrated care will make us more effective.*" (FG3, P1, 32-year-old male decision-maker) Such outcomes would make the integrated care package an attractive option for service provision:

"*Decision makers will not oppose if you mobilize and integrate resources and man power because you are going to minimize loss of resources and give care service for many patients.*" (K2)

Uniting the separate patient associations may also benefit in the same way: "*If the two or three come together, it will play a crucial role in terms of human resource utilization and knowledge transfer.*" (P6, FG3); "*If patients of all the three diseases come together and establish one large association, it will benefit all patients [. . .] If we integrate it, considering the resource, there is a chance we can get more things.*" (FG3, P1, 32-year-old male decision-maker).

Not all were optimistic or supportive of integration across the three diseases. Some were worried that integration may be detrimental in some regards, due to **competing priorities** when resources are limited, "*they may think there will be resource sharing . . .I also think that way.*" (K5). There were concerns that integrating care across the three diseases may **negatively affect uptake of care** by some with podoconiosis or LF, due to the **stigma** associated with leprosy.

"*If leprosy is integrated with the other disease, those patients taking podoconiosis and LF treatment could dropout.*" (K1);

"*From what I have seen they may refuse to take the treatment together because they think leprosy is communicable even by sitting together.*" (FG1, P2, 59-year-old male community representative)

Others were concerned that the current government health care strands could be problematic for successful integration: "*The problem in integrating leprosy treatment with others is the overall government structure problem and it is not designed in such a way.*" (K1). This was mostly related to separating leprosy from its current integration with tuberculosis: "*We shouldn't lose our experts. . .because, we have already integrated TB and leprosy and it has a good outcome. . . the one who is trained on TB is also trained on leprosy and it demands extra resources if we are trying to separate those providers or experts from the TB program.*" (K5). Another participant was worried that there were aspects of leprosy care that were not covered by the integrated package, such as ulcer management and specific medical treatment: "*In leprosy, the most probable disability is going to be neurological ulceration rather than lymphoedema.*" (K10)

This same participant expressed anxiety that integrated services rather than vertical ones **reduced expertise**: "*There are no vertical clinics in this village. These days every service is considered to be an integrated service* [. . .] *The health professionals don't have the ability to detect these diseases.*" (K10); and that they were already struggling to provide health care due to lack of resources and infrastructure: "*For treating podoconiosis patients, we even beg for materials from the cleaning staff. We can't even accommodate this kind of minor materials.*" As a result: "*I don't see the feasibility of the integration.*" (K10), even though they supported the ideal: "*Wound care clinics linked with psychiatric care and social work issues done by social workers need to be established*".

## Discussion

This qualitative study explored the feasibility, acceptability and appropriateness of implementing an integrated holistic care package for people with lower limb disorders caused by podoconiosis, LF or leprosy in a region of northwest Ethiopia where disease is highly prevalent. The EnDPoINT care package addresses three types of integration—integration into routine care; integration of mental health and psychosocial care (alongside limb care); and integration of care across the three diseases.

### Is it necessary and appropriate?

Our findings describe a significant burden of disease related to lower limb disorders including lymphoedema in the endemic districts of Ethiopia studied. This study found that individuals with lower limb disorders suffer a great deal, from the physical aspects of the disease as well as the psychosocial consequences with stigma and discrimination resulting in social isolation for affected persons and their families [20]. These lead to further consequences for their finances and health, in a vicious cycle. This burden–whether physical, psychological or financial–was similar regardless of which of the three diseases had caused the lower limb disorder, providing support for an integrated approach to providing care across the three diseases.

Like other studies, communities and individuals held beliefs about causation that hindered engagement with medical care and which perpetuated the stigma of affected individuals and their families. Consistent with other studies, experience and enactment of stigma was influenced by beliefs about the causes of lower limb lymphoedema, including ascribing the disease

to religious retribution, genetics and transmissible infection [21,22]. As elsewhere, this affected interpersonal relationships [23] including divorce and exclusion from social events. Our study highlights the absence of sufficient psychosocial support for affected persons within current care provision. This supports the appropriateness of an intervention that integrates mental health and psychosocial care as well as addressing stigma and lack of knowledge among those living in endemic areas [24].

Some of the challenges noted by our study participants was the spatial distribution of affected persons across large geographical areas in potentially hard-to-reach rural areas, and a general neglect for people affected by lower limb disorders within the health care system. This supports integration of services for lower limb disorders into routine care, particularly within primary health facilities and outreach in the community by health extension workers.

## Is it feasible?

Assessing feasibility of the program requires us to identify hurdles and how those can be overcome. Financial restraints, lack of material resources (including their supply chains and appropriate treatment spaces) and training of healthcare workers all need to be prioritised within current healthcare structures in order to facilitate the integration of services within routine care. Although there may be concerns that the package adds a burden of work, both integration of care across diseases and integration of holistic care (as opposed to vertical programmes) may improve the efficiency and cost-effectiveness of care, since resources–financial, material and human–are shared. Reassuringly, although the desire to provide a separate clinic to mitigate stigma would require extra space, this was not seen as a barrier of note by our participants. Collaboration between the health service, charities and community leaders was viewed constructively by our study participants, as urged by Engelman et al [25]; interestingly patient associations were seen as less helpful. However, some caution should be applied due to the limited inclusion of affected persons in this study; further exploration from their perspective on the role of patient associations would be helpful.

This study adds to the existing knowledge about barriers of care by highlighting the dearth of knowledge and experience amongst healthcare workers and care planners. Health care providers were inadequately prepared with either knowledge or resources to be able to manage people with lower limb disorders effectively. This is similar to a recent Rwandan study which demonstrated poor knowledge around podoconiosis as well as minimal clinical experience in treating affected persons [26]; this was compounded by shortages of supplies and drugs as in our study. Another recent study which explored the inclusion of podoconiosis in medical undergraduate studies across all endemic countries in Africa found there was insufficient specific teaching on podoconiosis [27], mirroring our respondents, along with earlier Ethiopia-specific reports [28] of high levels of misconceptions among health care professionals. Education interventions targeted to healthcare workers in endemic areas should be a component of any planned intervention to aid success. One advantage of implementing integrated programmes is that such interventions can be streamlined for training cohorts; within EnDPoINT, for example, health workers receive MMDP training for all three diseases combined as well as mental health training.

Vertical interventions requiring sufficient expertise to differentiate between the causative conditions may also be a barrier to care [29], and one of the benefits of our integrated care package is that it offers holistic treatment to people affected by lower limb disorders regardless of its cause.

## Is it acceptable?

Barriers to accessing and ongoing engagement with care are most commonly ascribed to stigma and financial reasons [30,31], however there are other important obstacles. Our study showed that making the intervention acceptable may require a reframing of what counts as "treatment". The materials as well as the clinical interaction suffer in affected persons' regard due to their intrinsic simplicity, which results in a failure to engage with care. Non-engagement (with a specific focus on non-use of footwear by people living in endemic areas) has also been explored [32,33], but less is known about other reasons for poor adherence. Misconceptions of care have been touched on elsewhere, such as in Tsegay et al's study in 2014 where disappointment with the simplicity of care on offer was related to discontinuing care [31]. Changing understanding of their importance and clinical value may facilitate better engagement with care by affected persons. Along with a focus on the "aetiology, preventability and treatability of the disease" [22], this could be included with the education and awareness initiatives. Our participants also felt that engagement would be improved by demonstrating success stories–actively through the use of expert patients [34,35]; and that failure to demonstrate a good response would have a negative effect, another "misconception of care" [30]. Setting realistic expectations for treatment response along with support by those who have been through the process themselves would be valuable in making the intervention acceptable to affected persons and their families.

In contrast to earlier reports of stigmatizing beliefs among healthcare workers [28], the attitudes of those interviewed in this study were much more positive about the conditions, recognised the stigma and were keen to work towards changing this. Participants were enthusiastic that an integrated program would provide tangible benefits for affected individuals and their families, while delivering better and more holistic care to a wider cohort of people living with lower limb disorders, done in a more efficient and cost-effective way.

However, there were also some concerns mentioned by our study participants, particularly in regard to integration of care for leprosy, since: leprosy programmes have traditionally been linked with those for tuberculosis in Ethiopia; leprosy can manifest itself in ways other than lower limb lymphoedema; and misinformation about the transmissibility of leprosy possibly limits acceptability of integrated services by people with podoconiosis and LF. Disentangling leprosy care from its current place within the TB programme may have negative sequelae, such as loss of expertise, and a requirement for extra resources, unless carefully managed. These kinds of challenges and concerns need to be addressed when planning and implementing integrated programmes. Interestingly, since completion of the EnDPoINT project, the Ethiopian Federal Ministry of Health has included leprosy into the priority NTD list /group in the 3rd National NTD Master Plan.

## Strengths and limitations of study

The inclusion of a variety of stakeholders from an endemic region, with the combined use of focus groups to stimulate discussion and in-depth interviews to probe in depth were strengths of this study. This study is part of a bigger project underpinned with implementation research concepts which will allow a wide range of mixed method approaches to explore fully the processes and outcomes involved in integrated care. Furthermore the inclusion of implementers and policy makers alongside other stakeholders as part of the consortium means that they have been kept abreast of research findings throughout the study and any concerns raised have been fed into later stages of the project.

There are limitations to this study. Firstly, this study was conducted in one area of northwest Ethiopia and the results may not be generalizable to other settings. Secondly, the data

comes from self-reporting rather than observed or witnessed behaviours. Thirdly, due to language limitations, one of the two researchers analysing data could not listen to the audio recordings; however, translations were done by the second researcher, who also conducted the interviews and contributed to data analysis; and we went back to the original data to settle queries about nuances. Due to the phased nature of the overall project, data collection in this study was limited by time constraints rather than primarily by data saturation.

An additional consideration is the limited inclusion of affected persons as participants. This was compounded by the merged nature of focus group 1, which held both affected persons and community members. While this could stimulate discussion, there is the risk that, despite the ground rules and the facilitators, affected persons may have felt restricted from fully sharing their experiences due to the presence of those by whom they may be stigmatized. In retrospect, including affected persons within the key informant interviews could have offset this, as well as recruiting more affected persons as participants.

The EnDPoINT care package was restricted to lower limb disorders and co-morbid mental health problems caused by podoconiosis, LF and leprosy. It may be interesting for future research to explore the feasibility and effectiveness of broadening the provision of integrated care to limb disorders caused by other non-communicable diseases, for example diabetic foot.

However, despite these limitations, the results provide support for the roll out of an integrated care package in the endemic areas of Ethiopia where the study was conducted.

## Conclusion

In conclusion, these findings support the introduction of an integrated holistic care package for lower limb disorders caused by podoconiosis, LF and leprosy in endemic areas in Ethiopia. Incorporating education of care givers and care recipients into any package would be essential as would prioritisation by decision-makers to enable allocation of time, resources and attention. It also suggests the need for further enquiry into factors that inhibit engagement with care in order to develop supportive strategies.

## Supporting information

**S1 Text. Topic guide questions.** English and Amharic versions of the questions used for the key informant interviews and focus group discussions.
(DOCX)

## Acknowledgments

The authors thank the other members of the EnDPoINT Research Team and Consortium for their input into the study, as follows: Tsige Amberbir, Vasso Anagnostopoulou, Hailom Banteyerga, Stephen A Bremner, Kebede Deribe, Abebaw Fekadu, Tanny Hagens, Damen Haile-Mariam, Natalia Hounsome, Louise A Kelly-Hope, Hayley MacGregor, Henock B Taddese, Tadesse Tesfaye, Seifu Tirfie, and Abebayehu Tora. They also thank Clare Callow for her project management of the NIHR grant through which EnDPoINT is funded, as well as Tesfaye Asefa, Bethelhem Fekadu, Grit Gansch, Samrawit Ketema for their support with the administrative aspects of the study. Our sincerest thanks go to all participants of the EnDPoINT study.

## Author Contributions

**Data curation:** Abraham Tesfaye.

**Formal analysis:** Bethany Davies, Mersha Kinfe.

**Funding acquisition:** Gail Davey.

**Investigation:** Mersha Kinfe, Oumer Ali, Asrat Mengiste, Abraham Tesfaye, Mossie Tamiru Wondimeneh, Gail Davey, Maya Semrau.

**Methodology:** Abraham Tesfaye, Gail Davey, Maya Semrau.

**Project administration:** Mersha Kinfe, Abraham Tesfaye.

**Resources:** Gail Davey.

**Supervision:** Gail Davey.

**Writing – original draft:** Bethany Davies.

**Writing – review & editing:** Bethany Davies, Mersha Kinfe, Oumer Ali, Asrat Mengiste, Abraham Tesfaye, Gail Davey, Maya Semrau.

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
