## [Decision Letter · Decision Letter 0]

20 Sep 2021

Dear Dr Davies,

Thank you very much for submitting your manuscript "Stakeholder perspectives on an integrated package of care for lower limb lymphoedema of varying causes: a qualitative study" for consideration at PLOS Neglected Tropical Diseases. As with all papers reviewed by the journal, your manuscript was reviewed by members of the editorial board and by several independent reviewers. The reviewers appreciated the attention to an important topic. Based on the reviews, we are likely to accept this manuscript for publication, providing that you modify the manuscript according to the review recommendations. 

Sincerely,

Alison Krentel

Associate Editor

Godfred Menezes

Deputy Editor

Thank you for your submission. Please consider the reviewers' comments as you prepare the revision of this paper.

Reviewer's Responses to Questions

**Key Review Criteria Required for Acceptance?**

**Methods**

-Are the objectives of the study clearly articulated with a clear testable hypothesis stated?

-Is the study design appropriate to address the stated objectives?

-Is the population clearly described and appropriate for the hypothesis being tested?

-Is the sample size sufficient to ensure adequate power to address the hypothesis being tested?

-Were correct statistical analysis used to support conclusions?

-Are there concerns about ethical or regulatory requirements being met?

Reviewer #1: The principle behind this study is sound and practical and the potential benefits undoubted

I had some difficulty disentangling the physical abnormalities associated with the three target diseases – podoconiosis and LF whose main pathological focus is lower limb oedema ( it may be wider than this in LF) and leprosy where the main damage is neuropathic resulting in limb ulceration with surrounding inflammation in some cases leading to localised oedema. So isn’t this study focussing on the impact and resolution of lower limb disabilities rather than lymphoedema per se ?

Rather along the same lines I wonder if stigma is based on disease specific beliefs rather than syndrome specific beliefs. Certainly other studies have shown that leprosy carries very specific associations with mental and social consequences - even without limb ulceration. Does this apply to the other diseases ? If so how does this affect the questions and their analysis

How do other diseases including NCDs fit in with this package. I realise that idiopathic or cancer related lymphoedemas are probably uncommon in this setting. But what about diabetic foot ?

Reviewer #2: The methods are well written and relevant to the purpose of the study described. However, I would recommend a few clarifications or alterations prior to publication as follows: 

Could better description of the MRC framework on complex interventions and the CICE framework be provided, with a particular focus on how the endpoint study and this work map to the framework. This could perhaps be provided through a figure?

The authors described that the number of participants recruited is based on a data saturation approach- yet numbers seem small for data saturation to have been achieved- particularly given the diversity of stakeholders included in the study. Could this be further clarified or amended?

Line 150: The authors state 'all participants were encouraged to contribute and be heard- Can the authors expand on how this was done?

Ethics- merged FGDs with community members and people affected- how was stigma/confidentiality etc. managed? is it possible that affected persons felt unable to share their true experiences if those who may stigmatise were also part of the group discussion- could this be discussed further? 

Further, where issues of mental distress may have been disclosed by participants how was this managed and supported?

Line 181-184 belongs in methods otherwise reader left guessing on group composition in the methods

Reviewer #3: Clear methods, well crafted description of rationale and background

**Results**

-Does the analysis presented match the analysis plan?

-Are the results clearly and completely presented?

-Are the figures (Tables, Images) of sufficient quality for clarity?

Reviewer #1: Can you break down the groups studied further for instance to include some numbers eg how many lymphoedema patients were in the stakeholder group. 

Although this is explained in the supplementary data it would be useful to indicate in the text if the questions were asked were about each disease or the syndrome ie a disabled or swollen limb.

Please expand on the costs of joining a patients’ organisation . Can you give an approximate figure ?

Reviewer #2: The results match the analysis plan well, however, a little more clarity could be provided in the results to support the reader to follow clearly.

In Table One- as well as providing a summary of theme titles, could a summary of the content of each theme also be provided- this would help anchor the reader as they move through the results text. Otherwise this is relatively long and easy to get lost. 

Could quotes that do not flow directly in the text also be indented. Currently, the text flows as one long narrative in places which makes it hard for the reader to differentiate between the analytical point presented in the results and the supporting evidence. Perhaps sub-sub themes could also be bolded when referred to to make this clear to the reader. 

Matter matters- could this theme heading be adjusted it's a bit confusing- perhaps material matters would be a better term?

Reviewer #3: Inclusion of the free text of some of the responses whilst increasing the length of the paper provides a richness and widespread applicability which other disease groups might find informative.

**Conclusions**

-Are the conclusions supported by the data presented?

-Are the limitations of analysis clearly described?

-Do the authors discuss how these data can be helpful to advance our understanding of the topic under study?

-Is public health relevance addressed?

Reviewer #1: Can you indicate how this will translate into addressing the concerns raised by the various groups ?

Reviewer #2: The conclusions are largely supported by the data, however, some of the less positive points related to service integration could be further emphasised, particularly related to stakeholders concerns related to integration including limited availability of space to provide a separate clinic but a wish to do so to mitigate stigma; challenges with shifting leprosy from TB service integration; and overburdening of health staff. 

The limitations could also further emphasise lack of direct (non-group specific) involvement of affected persons- their inclusion in whole group discussions may have affected their responses- or some discussion of how this was mitigated should be provided.

Reviewer #3: The conclusions are clear and recommendations useful

**Editorial and Data Presentation Modifications?**

Reviewer #2: - Line 69- (world bank)- needs removing

- Line 119- Deribe PLS Map Model- needs reference

- Line 120 (census)- needs removing

Reviewer #3: accept

**Summary and General Comments**

Reviewer #2: This paper is well written and a needed contribution to highlight the importance of health system stakeholder involvement in the design of integrated interventions for lower limb lymphedema care. The study has many strengths and will provide learning to other settings. As well as suggestions above, a few additional points for consideration in altering the manuscript prior to publication as follows: 

- Could the difference between mental health and psychosocial care be better articulated in the abstract and introduction where this is referred to as two separate components?

- Could the closing paragraph of the introduction include a more specific section on the purpose of this specific study rather than the overarching endpoint work?

- Could you define how you have conceptualised appropriateness, feasibility and acceptability either within your introduction or at the beginning of your discussion?

- It is great that affected persons were included in group discussions. However, it would be good to see more about the strengths and limitations of the approach in including them with other community members and how far the data collated is a true representation of their views. I would also urge caution with interpretation of the finding around the role of patient associations in integrated care- from your data- my understanding is that this view point comes from health stakeholders not patients themselves. Perhaps this should be framed as a findings that needs further interrogation from the perspective of patients?

Overall, a good paper and i look forward to seeing the outputs of the rest of the Endpoint intervention.

Reviewer #3: This is a well written and original paper which has potential for widespread applicability in the space of integrated approaches for NTDs

PLOS authors have the option to publish the peer review history of their article (what does this mean?). If published, this will include your full peer review and any attached files.

Reviewer #1: No

Reviewer #2: No

Reviewer #3: Yes: Dr L Claire Fuller

Figure Files:

Data Requirements:

Reproducibility:

References

---

## [Editor Report · Decision Letter 1]

4 Jan 2022

Dear Dr Davies,

We are pleased to inform you that your manuscript 'Stakeholder perspectives on an integrated package of care for lower limb disorders caused by podoconiosis, lymphatic filariasis or leprosy: a qualitative study' has been provisionally accepted for publication in PLOS Neglected Tropical Diseases.

Best regards,

Alison Krentel

Associate Editor

Godfred Menezes

Deputy Editor

Thank you for the careful revisions that you have done to the paper and your detailed responses to the comments from the reviewers. Your explanations related to changes you made and changes you were unable to make were noted. We are comfortable with the amendments that have been made to the manuscript and I am pleased to recommend this paper for publication.

---

## [Editor Report · Acceptance letter]

17 Jan 2022

Dear Dr Davies,

We are delighted to inform you that your manuscript, "Stakeholder perspectives on an integrated package of care for lower limb disorders caused by podoconiosis, lymphatic filariasis or leprosy: a qualitative study," has been formally accepted for publication in PLOS Neglected Tropical Diseases.

Best regards,

Shaden Kamhawi

co-Editor-in-Chief

Paul Brindley

co-Editor-in-Chief
